# Causal Foundation Models for Time Series based on Prior-Data fitted Networks

**Dennis Thumm** [* 1]  **Arik Reuter** [* 2 3]  **Jake Robertson** [* 4 5]  **Shi Bin Hoo** [5 4]  **Adrian Weller** [2 6]  **Frank Hutter** [4 5 7]
**Ying Chen** [1]  **Bernhard Schölkopf** [3 7]

## Abstract

Foundation models trained on synthetic data have recently transformed tabular machine learning and time series forecasting, with Prior-Data Fitted Networks (PFNs) being the leading approach. Causal Foundation Models (CFMs) extend the PFN paradigm from prediction to causal effect estimation. However, existing CFMs are limited to static (i.i.d.) settings and do not address the challenges posed by temporal data. In this work, we extend the PFN framework to estimate causal effects in time series. We identify prior-design criteria and introduce a novel temporal causal PFN prior that yields data with meaningful causal effects while maintaining the stability of sampled trajectories. Furthermore, we provide theoretical results that shed light on the asymptotic behaviour of the conditional interventional distribution implied by a temporal causal prior. Empirically, we find that PFNs can successfully learn to estimate causal effects in temporal settings, achieving promising performance on synthetic data covering the back-door and front-door settings.

## 1. Introduction

Understanding the causal effect of interventions in time series is central to scientific and decision-making problems across domains such as healthcare, finance, economics, and climate science. This problem can be framed as *interventional time series forecasting*, where the goal is to answer the question: "*what happens to a system if we intervene at a particular point in time?*" Classical approaches to causal

inference in time series problems rely on carefully specified structural assumptions, such as knowledge of the temporal causal graphs or sequential ignorability (Pearl, 2009; Runge et al., 2023; Peters et al., 2017). Other methods make parametric assumptions and are tailored to specific regimes, such as linear models and low-dimensional settings (Hyvärinen et al., 2008; Runge et al., 2023), all of which can limit their applicability in complex real-world time series contexts.

In the domains of tabular machine learning and time series forecasting, the dependence on manual model selection has recently been reduced through the use of Foundation Models in the form of Prior-Data-Fitted Networks (PFNs; Müller et al. (2021)). Causal, tabular and time series Foundation Models are trained once on large collections of data and adapt their predictions in-context to new tasks, removing the need to design bespoke estimators for each problem, and achieve state-of-the-art predictive performance (Hollmann et al., 2025; Qu et al., 2026; Ansari et al., 2025b).[1] This raises a natural question: *Can PFNs serve as an effective approach for predicting causal effects in time series?*

In this work, we take a first step toward answering this question by investigating PFN-based causal foundation models for the problem of interventional time series forecasting. Please refer to Figure 2 in Appendix A.1 for an overview over the proposed methodology.[2]

## 2. Methodology: Prior fitting for Causal Effect Estimation in Time Series

**Temporal Causal Structures**   A canonical way to formalize causality in time series is via Temporal Structural Causal Models (TSCMs; Peters et al., 2017; Boeken and Mooij, 2024; Thumm and Chen, 2026). More specifically, let $((a_t, \mathbf{x}_t, y_t))_{t=0}^{T_{max}}$ denote a (multivariate) time series ranging from time $t = 0$ to time $t = T_{max}$. Here, $a_t \in \mathbb{R}$ denotes a treatment variable at time $t \in \mathbb{N}_0$, $\mathbf{x}_t \in \mathbb{R}^{N-2}$ denotes the (potentially multivariate) covariates, and $y_t \in \mathbb{R}$

[1]Department of Mathematics, National University of Singapore [2]Department of Engineering, University of Cambridge, UK [3]Max Planck Institute for Intelligent Systems, Tübingen, Germany [4]Prior Labs, Freiburg, Germany [5]Department of Computer Science, Albert-Ludwigs-Universität Freiburg, Germany [6]Alan Turing Institute, London, UK [7]ELLIS Institute Tübingen, Germany. Correspondence to: Dennis Thumm <dennis.thumm@u.nus.edu>.

*Proceedings of the $2^{nd}$ ICML Workshop on Foundation Models for Structured Data*, Seoul, South Korea. 2026. Copyright 2026 by the author(s).

---

[1]We discuss related work in terms of traditional methods for causal discovery and causal inference, Tabular and Time Series Foundation Models, as well as Causal Foundation Models in Appendix B.

[2]We release our code here: https://anonymous.4open.science/r/do-over-time-pfn-EF8D

the outcome. The total number of variables considered is thus $N$. Ultimately, the goal is to predict the outcome $y_T$ when performing an intervention on $a_T$ taking into account past values for $y_t$, $a_t$ and $\mathbf{x}_t$. A TSCM describes the causal data-generating process underlying the time series. Following Thumm and Chen (2026), a TSCM $\psi$ is defined as a triplet $\psi = (\mathcal{G}, \mathbf{F}, P^\epsilon)$ that contains a time-lagged DAG $\mathcal{G} := (G_0, G_1, \ldots, G_K)$, where $G_0 \in \mathbb{R}^{N \times N}$ encodes instantaneous edges, and $G_k \in \mathbb{R}^{N \times N}$ edges from time point $t - k$ to time $t$, $k = 1, \ldots, K$; i.e. $G_k$ determines which variables $k$ time-steps in the past have direct effects on the present. Furthermore, $\mathbf{F} = (f_1, f_2, \ldots, f_N)$ contains the mechanisms for each variable. Let $z_{i,t}$, $i = 1, \ldots, N$ denote a variable at time $t$ (which can be a treatment, covariate, or outcome), such that

$$z_{i,t} = f_i \left( \mathrm{Pa}_0(z_i), \mathrm{Pa}_1(z_i), \ldots, \mathrm{Pa}_K(z_i) \right) + \epsilon_{i,t}, \quad (1)$$

where $\mathrm{Pa}_k(z_i)$ denotes the parents of the $i$-th variable in $G_k$, and $\epsilon_{i,t}$ are sampled independently from $P^\epsilon$. Furthermore, Equation 1 directly prescribes how to sample observational data from the TSCM via ancestral sampling, and we denote a slice of an observational dataset sampled in this way from time $T_0$ to $T_1$ as $\mathcal{D}_{T_0}^{T_1} = ((a_t, \mathbf{x}_t, y_t))_{t=T_0}^{t=T_1}$.

**Interventions and Interventional Distributions**  In addition to sampling observational data, TSCMs allow us to formalize interventions. In the present work, we consider hard interventions that remove all incoming edges into a variable $a_T$, at a specific time-point $T$. The effect of this intervention in a given TSCM $\psi$ can be expressed via the interventional distribution $p(y_T|do(a_T), \psi)$. By assuming a prior $p(\psi)$ over TSCMs, we can obtain the following conditional interventional distribution (CID), which is the central object we want to infer throughout this paper:

$$p(y_T|do(a_T), \mathcal{D}_0^{T-1}) =$$
$$\int p(y_T|do(a_T), \mathcal{D}_{T-k}^{T-1}, \psi) p(\psi|\mathcal{D}_0^{T-1}) d\psi. \quad (2)$$

This CID answers the question "*Which value will the outcome variable y take at time-step T, provided we intervene on variable a at time T, while taking into account past values for each variable?*"

The CID in Equation 2 can be approximated using PFNs, which we explain in more detail in Appendix C.

### 2.1. Implementing a Prior for Causal Effect Estimation on Time Series

Our prior is a vectorized sampler over temporal structural causal models (TSCMs) covering the two canonical identification structures—back-door and front-door (with a hidden confounder)—each instantiated by a fixed per-structure

DAG builder with random mechanism weights. Every sampled TSCM specifies linear-plus-nonlinearity dynamics with VAR($L \leq 5$) temporal memory, additive Gaussian process noise, and a randomly drawn pointwise activation per variable. Crucially, the mechanism draw is shared between the paired observational rollout $X_b^{\mathrm{obs}}$ and interventional rollout $X_b^{\mathrm{int}}$ of each sample, so that the only systematic difference between the two trajectories is the do-intervention itself—a property we find necessary for a non-collapsing training signal. Stability is enforced by a per-SCM spectral-radius clip on the reduced-form VAR($L$) companion matrix; together with strictly positive-variance Gaussian innovations and a bounded element-wise nonlinearity, this guarantees that the resulting Markov chain is geometrically ergodic and positive Harris-recurrent, which is the regularity condition our convergence theorem relies on. We use two concrete calibrations throughout the paper: an *oscillatory hardened* (OSC) prior ($\rho_{\max}=0.95$, $\sigma_{\mathrm{noise}}=0.05$) that produces a strong, long-horizon causal signal, and a *break-trajmean* (BTM) prior ($\rho_{\max}=0.70$, $\sigma_{\mathrm{noise}}=0.10$) that trades off autocorrelation strength for a stricter benchmark in which the trivial per-trajectory pre-intervention mean no longer ties the trained PFN on RMSE. We refer the details of our prior-design to Appendix E.1

## 3. Convergence of the Conditional Interventional Distribution

When designing a prior $p(\psi)$ over TSCMs that induces a conditional interventional distribution $p(y_T|do(a_T), \mathcal{D}_0^{T-1})$ (as explained in Section 2), it is non-trivial to describe how this posterior behaves asymptotically—yet, asymptotic properties are widely considered critical in causality since agreed-upon real-world benchmarks do not exist. The question of asymptotic behaviour becomes (especially) meaningful when assuming a ground-truth TSCM $\psi_0$ that generates all data, such that we can state the frequentist goal of "recovering" $p(y_T|do(a_T), \mathcal{D}_0^{T-1}) \approx p(y_T|do(a_T), \mathcal{D}_{T-k}^{T-1}, \psi_0)$ for large $T$. Even though all SCMs $\psi' \in [\psi_0]$ in the observational equivalence class of $\psi_0$ are indistinguishable from $\psi_0$ based on $\mathcal{D}_0^{T-1}$, they might still yield different causal effects. This is reflected in the convergence of $p(y_T|do(a_T), \mathcal{D}_0^{T-1})$ to $p(y_T|do(a_T), \mathcal{D}_{T-k}^{T-1}, [\psi_0])$, i.e, *convergence up to observational equivalence*, where we define

$$p(y_T|do(a_T), \mathcal{D}_{T-k}^{T-1}, [\psi_0]) :=$$
$$\int p(y_T|do(a_T), \mathcal{D}_{T-k}^{T-1}, \psi') p(\psi'|[\psi_0]) d\psi', \quad (3)$$

and $p(\psi'|[\psi_0])$ directly results by the notion of observational equivalence relative to the prior $p(\psi)$:

**Theorem 3.1.** *Let $\psi_0$ denote a TSCM generating the observational data $\mathcal{D}_0^{T-1}$, and fix the values for $y_T$, $a_T$, as well as $\mathcal{D}_{T-k}^{T-1}$, as $T$ grows. Further denote the observational equiv-*

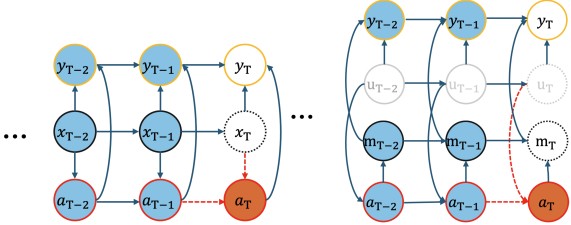

*Figure 1.* Identification settings for our experiments: **Left**: Back-Door case, and **right** Front-Door case. Both case-studies assume autoregressive edges for all variables.

alence class of $\psi_0$ by $[\psi_0]$, s.t. $p(\mathcal{D}_0^{T-1}|\psi') = p(\mathcal{D}_0^{T-1}|\psi_0)$ for all $\psi' \in [\psi_0]$ and all $T \in \mathbb{N}_0$. Then

$$p(y_T|do(a_T), \mathcal{D}_0^{T-1}) \longrightarrow p(y_T|do(a_T), \mathcal{D}_{T-k}^{T-1}, [\psi_0]) \quad (4)$$

in $P_{\psi_0}$-probability for $T \to \infty$. Here we assume standard regularity conditions—Polish-space variables admitting densities, a positive Harris recurrent observational chain (*Meyn and Tweedie, 2012*), a measurable observational-equivalence map, a bounded continuous interventional functional, and standard posterior-consistency assumptions on the prior over transition densities; the full statement and proof are in Appendix D.

The proof and a more formal version of this theorem can be found in Appendix D.

## 4. Experiments

We empirically investigate the previously unexplored task of prior-data fitting for estimating causal effects from time series in simple, controlled setups that encompass classical Back-Door and Front-Door setting (See Figure 1). We train a transformer with a 6-layer encoder, a cross-variable mixer with 3 cross-attention blocks, gated residuals, and a quantile head on synthetic data sampled from those graph-structures according to our temporal causal prior (Section 2.1). This model is referred to as TrainedPFN_int. An important baseline is a version of our model that is trained on observational data from the same prior but is otherwise completely identical TrainedPFN_obs. Please refer to Appendix G for more details on the architecture and training.

### 4.1. Per-Structure Training

First, we train a separate PFN for each individual case–study and compare to classical and foundation-model-based baselines that use the correct adjustment formulas.

**Baselines.** We compare the trained PFN against several classes of baselines designed to probe different aspects of the problem. First, we include two simple trajectory-level heuristics: TRAJMEAN, predicting the pre-intervention trajectory mean of the query variable, and AR1, which predicts

*Table 1.* Baseline comparison at $T$=1,000 on the oscillatory prior (matched 320-query set, un-normalized RMSE/direction-accuracy; $\pm$ are $B$=200 bootstrap SEs). Bold/underline mark the best/second-best per column; evaluation protocol in Appendix J.

| Method | BD | | FD | |
| --- | --- | --- | --- | --- |
| | RMSE $\downarrow$ | Dir. acc. $\uparrow$ | RMSE $\downarrow$ | Dir. acc. $\uparrow$ |
| TrainedPFN_int | $\underline{0.70} \pm 0.06$ | $75.9\% \pm 3.1$ | $\mathbf{0.45} \pm \mathbf{0.05}$ | $\underline{82.8\%} \pm 2.8$ |
| TrainedPFN_obs | $0.78 \pm 0.10$ | $61.3\% \pm 3.5$ | $0.55 \pm 0.07$ | $76.1\% \pm 3.1$ |
| TRAJMEAN | $0.71 \pm 0.07$ | $74.3\% \pm 3.2$ | $\underline{0.46} \pm 0.05$ | $81.9\% \pm 2.9$ |
| AR1 | $0.98 \pm 0.11$ | $46.6\% \pm 3.6$ | $1.04 \pm 0.12$ | $51.4\% \pm 3.7$ |
| *Classical structural baselines* | | | | |
| VAR($p$=3)_int | $0.78 \pm 0.13$ | $\underline{88.0\%} \pm 2.4$ | $0.46 \pm 0.06$ | $81.9\% \pm 2.9$ |
| VAR($p$=3)_obs | $0.77 \pm 0.09$ | $\underline{70.2\%} \pm 3.3$ | $0.59 \pm 0.08$ | $80.8\% \pm 2.9$ |
| BD-OLS_int | $0.79 \pm 0.12$ | $\underline{88.0\%} \pm 2.4$ | — | — |
| BD-OLS_obs | $0.75 \pm 0.09$ | $67.5\% \pm 3.4$ | — | — |
| *Foundation-model adjustment baselines* | | | | |
| TabPFN-Adj_int | $\mathbf{0.49} \pm \mathbf{0.05}$ | $86.9\% \pm 2.4$ | $0.66 \pm 0.07$ | $77.0\% \pm 3.1$ |
| TabPFN-Adj_obs | $0.75 \pm 0.09$ | $72.8\% \pm 3.2$ | $0.55 \pm 0.06$ | $\mathbf{84.2\%} \pm \mathbf{2.7}$ |
| Chronos-Adj_int | $0.71 \pm 0.12$ | $\mathbf{88.0\%} \pm \mathbf{2.4}$ | $0.86 \pm 0.08$ | $71.0\% \pm 3.4$ |
| Chronos-Adj_obs | $0.75 \pm 0.08$ | $75.4\% \pm 3.1$ | $0.66 \pm 0.10$ | $77.6\% \pm 3.1$ |

the last observed value. Second, we consider *classical structural baselines* that explicitly implement the appropriate adjustment formulas for each causal graph. These include multivariate autoregressive forecasting via VAR($p$=3) and back-door adjustment using ordinary least squares (BD-OLS). Third, we evaluate *foundation-model adjustment baselines* built on top of pretrained predictive foundation models, namely TABPFN-2 (Hollmann et al., 2022) and CHRONOS-2 (Ansari et al., 2025b). These baselines serve as approximate "oracle" methods in the sense that they are given the correct causal adjustment strategy. Furthermore, we differentiate between those actually causal baselines (indicated by _int_), and their purely observational counterparts (indicated by _obs_).

**Comparison against baselines.** Table 1 compares the trained PFN against classical causal baselines and adjustment procedures implemented using pretrained foundation models. Overall, the results demonstrate that prior-data fitting enables transformers trained under implicit graphical assumptions to recover meaningful interventional distributions, producing similar results as compared to graph-informed adjustment across a range of classical identification settings.

In the **back-door** setting, the trained PFN substantially outperforms its obs-only counterpart in both RMSE (0.70 vs. 0.78) and direction accuracy (75.9% vs. 61.3%), indicating that causal pretraining allows the model to leverage interventional information beyond observational correlations alone. While explicit adjustment baselines based on pre-trained foundation models, TabPFN-Adj_int and Chronos-Adj_int, achieve the strongest overall performance in this setting, these methods are additionally provided with the correct adjustment strategy at inference time. In contrast,

*Table 2.* Joint (ALL) vs. per-structure (PER) trained PFN at $T{=}1{,}000$, $n{=}320$, bootstrap SE $B{=}200$. PER columns are matched train/eval priors; ALL columns are evaluated on each structure's canonical query offset range. Bold indicates the best value per row.

| | | Per-structure | | All-structure | |
|---|---|---|---|---|---|
| Prior | Case | RMSE ↓ | Dir. acc. ↑ | RMSE ↓ | Dir. acc. ↑ |
| *Causal models* | | | | | |
| OSC | BD | **0.69 ± 0.06** | 75.9% ± 3.1 | 0.73 ± 0.05 | **77.5% ± 3.0** |
| OSC | FD | 0.45 ± 0.05 | 82.8% ± 2.8 | 0.45 ± 0.05 | **83.0% ± 2.8** |
| BTM | BD | 1.01 ± 0.09 | 72.6% ± 3.2 | 1.01 ± 0.09 | **73.1% ± 3.2** |
| BTM | FD | 0.42 ± 0.03 | **87.0% ± 2.5** | 0.42 ± 0.03 | 86.9% ± 2.5 |
| *Observational-only models* | | | | | |
| OSC | BD | 0.78 ± 0.10 | 61.3% ± 3.5 | 0.78 ± 0.10 | 61.3% ± 3.5 |
| OSC | FD | 0.55 ± 0.07 | **76.1% ± 3.1** | 0.54 ± 0.07 | 74.7% ± 3.2 |
| BTM | BD | 1.05 ± 0.12 | **66.5% ± 3.4** | 1.05 ± 0.12 | 66.0% ± 3.4 |
| BTM | FD | 0.48 ± 0.03 | 83.4% ± 2.8 | **0.43 ± 0.03** | **89.6% ± 2.3** |

the trained PFN needs to learn this behaviour during pre-training.

In the **front-door** setting, the trained PFN achieves the best RMSE overall ($0.45 \pm 0.05$), outperforming all explicit adjustment baselines. Moreover, it improves substantially over the obs-only variant ($0.55 \pm 0.07$), suggesting that the model successfully captures aspects of the more challenging front-door structure from synthetic causal pretraining alone. Interestingly, even simple trajectory statistics already provide a strong signal under the oscillatory prior, making this a surprisingly competitive setting for heuristic baselines.

### 4.2. Joint Multi-structure vs. Per-Structure Training

A natural follow-up question is whether a *single* PFN trained jointly across all identification settings can match the performance of the per-structure specialists, providing empirical evidence towards the ability of the model to generalize. To investigate this, we train two additional models per prior, one causal and one observational-only, by jointly sampling from the Back-Door, Front-Door, and Instrumental Variable (IV)[3] settings during training, where IV enriches the training distribution. We then evaluate these joint models separately on each individual structure.

**Comparison against per-structure training.** Table 2 shows that joint training across multiple causal identification settings is highly effective. Across almost all settings, the jointly trained models match the performance of the corresponding per-structure specialists, despite sharing a single set of parameters across fundamentally different causal graph structures. Importantly, these gains are achieved without sacrificing performance on either the Back-Door or Front-Door tasks, where the jointly trained models remain nearly identical to the specialized models. In

some cases, joint training even improves observational-only models, most prominently for the BTM Front-Door setting, where directional accuracy increases from $83.4\%$ to $89.6\%$ while simultaneously reducing RMSE. Overall, these results indicate that prior-data fitting naturally supports a unified multi-structure training paradigm: a single transformer can amortize inference across multiple causal identification settings while retaining the performance of specialized models trained on only one structure.

### 4.3. Ablations

We complement the per-structure and joint training results above with three ablations whose detailed tables are deferred to Appendix H.

**Lagged effects.** Across the 12 checkpoints obtained by crossing the three identification structures with {no-lag, lag} and {causal, obs-only} training, every causal model attains positive $R^2$, and the causal-over-observational advantage is largest for FD-lag ($84.6\%$ vs. $53.5\%$ direction accuracy) and small but consistent for BD, while IV is essentially tied between the two training regimes (Table 6).

**Trajectory length.** Sweeping the evaluation horizon over $T \in \{100, 500, 1{,}000, 2{,}000, 5{,}000\}$ on the six causal checkpoints, the model generalises across a $50\times$ range with $R^2 > 0.4$ in every cell, even though training samples never exceed $T{=}2{,}000$ (Table 7).

**OOD prior generalization.** We cross-evaluate the OSC- and BTM-trained checkpoints against *both* prior distributions to quantify how much of the model's competence is specialised to its training prior (Table 9).

## 5. Conclusion

In this work, we take the first step towards Prior-Data-Fitted Networks as a generic method for predicting causal effects in time series. We propose a suitable temporal PFN prior, discuss the resulting asymptotics, and empirically show that prior-fitting can be an effective method.

**Limitations and outlook.** First, our experiments are restricted to synthetic data generated from relatively small temporal causal graphs. While this allows for controlled analysis, it remains unclear how well such models generalize to large-scale, noisy, and nonstationary systems. While we perform an out-of-distribution (OOD) analysis, designing priors that guarantee good real-world performance requires realistic causal benchmarks, which are currently lacking (Poinsot et al., 2025). Finally, we consider hard interventions at a single time-step as a simple yet meaningful starting point. However, some applications require reasoning about sustained interventions or multi-step forecasts.

---

[3]See Figure 3 in Appendix A.1 for this structure.

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

# A. Explanatory Figures

## A.1. Methodology

Figure 2 provides an overview over the proposed methodology: Prior-fitting for causal effect estimation in time series.

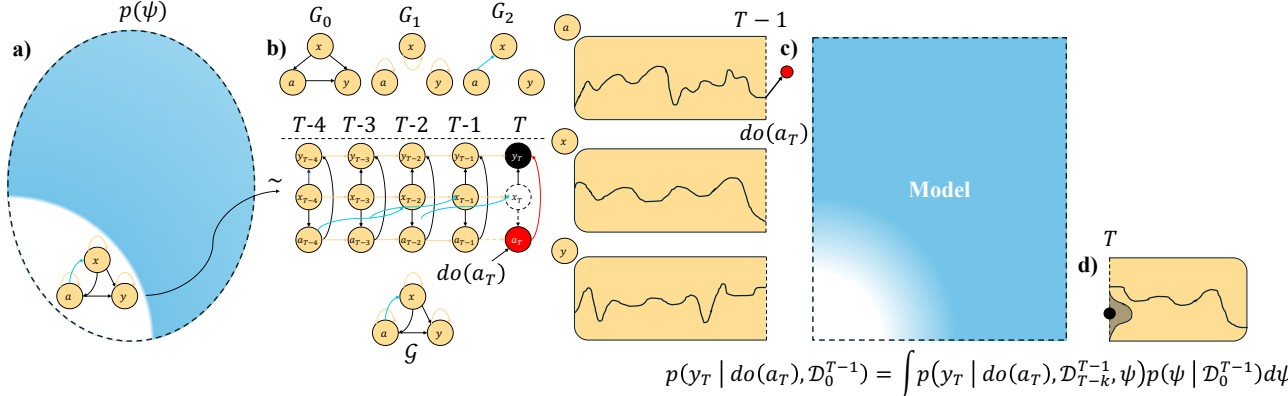

*Figure 2.* We consider the problem of causal effect estimation in time series. Our pre-training loop follows the above procedure. **(a)** A Temporal Structural Causal Model (TSCM) $\psi$ is sampled from our prior over TSCMs $p(\psi)$, generating **(b)** an unrolled causal structure dictated by $\mathcal{G} = \{G_0, G_1, ..., G_k\}$ which can contain complex system-level causal dynamics. **(c)** We perform an intervention $do(a_T)$ on treatment variable $a$ at a discrete time step $T$ and **(d)** predict the interventional outcome $y_T$ followed by loss computation and a gradient step. Repeating those steps trains our model to approximate the Conditional Interventional Distribution (CID) for time series $p(y_T|do(a_T), \mathcal{D}_0^{T-1})$ which integrates over potential causal explanations of the observed dynamics of the system. At inference-time, the pre-trained model takes an observational time series $\mathcal{D}_0^{T-1}$, together with an interventional query-value $a_T$ as input and predicts $p(y_T|do(a_T), \mathcal{D}_0^{T-1})$.

## A.2. Instrumental Variable

Training on BD, FD, IV but evaluating on BD, FD treats IV as a training-time auxiliary task in Section 4.2 rather than a deliverable to benefit regularisation and inductive bias.

**Identification-logic dichotomy.** Since FD and IV occupy the same identification regime under the trained PFN's own probe, evaluating on BD, FD already covers both qualitative routes and IV evaluation is informationally redundant with FD. Including IV in training but not evaluation reframes it as an auxiliary task that gives the model *more examples* of post-intervention-trajectory identification, while we test whether that lesson transferred on the cleaner FD case.

**Capacity-preservation test.** The interesting question for any foundation-model claim is whether multi-structure training compromises performance on the structures with strong baselines and well-characterized theory. BD and FD have clean adjustment formulas (BackDoorOLS, FrontDoorOLS) and prior CFM work (Balazadeh et al., 2025; Ma et al., 2025; Robertson et al., 2025) is centred on these two regimes. Evaluating the all-structure model on BD, FD asks answers: did exposure to IV during training degrade the canonical cases? A null result is itself a strong claim, adding identification structures to the training mix is a free lunch.

# B. Related Work

To contextualize our work, we begin by briefly discussing traditional methods for causal discovery and causal inference in time series problems, followed by detailing recent advances in tabular and time series Foundation Models, including Causal Foundation Models.

## B.1. Causal discovery and inference in time series.

To estimate causal effects in time series, one would typically start by inferring the underlying temporal causal structure, e.g., formalized via a TSCM, introduced in Section 2. In contrast to the i.i.d setting, the temporal order provides a natural constraint on some of the variables—effects cannot precede their causes. In case the ground-truth causal structure is not

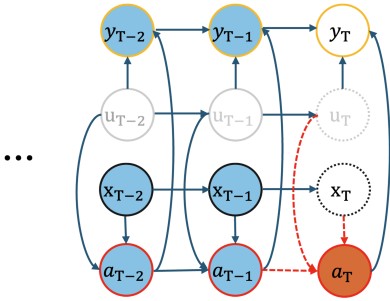

*Figure 3.* Instrumental Variable (IV) case with instrument $x$, unobserved confounder $u$, outcome $y$ and intervention variable $a$. In all cases, we predict $p(y_T|do(a_T), \mathcal{D}_0^{T-1})$, where $x_T$ and $m_T$ are marginalized out. All case-studies assume autoregressive edges for all variables.

known from domain knowledge, causal discovery algorithms are necessary. Standard procedures based on conditional independence tests (Spirtes et al., 2000) have been adapted to the time series case (Runge, 2020; Gerhardus and Runge, 2021). Granger Causality is a (historically) especially popular concept in the context of causal discovery in time series, and can also be framed as a procedure based on conditional independence tests (Granger, 1969; Peters et al., 2017). Furthermore, several methods introduced by Hyvärinen et al. (2008); Peters et al. (2013); Wu et al. (2022) are examples of causal discovery algorithms using restrictions on functional forms and/or noise-distributions for causal discovery in time series. After a temporal causal structure has been identified, one can proceed analogously to the "normal" i.i.d. case by determining an adjustment formula (if it exists), a causal estimate and ultimately carrying out the necessary estimation (Pearl, 2009). However, the estimation step can be more difficult (or even impossible) depending on the temporal dynamics of available data in comparison to the i.i.d. case. We refer to Runge et al. (2023) for a recent overview of traditional methods for causality in time series. In contrast to those multi-step methods, the PFN-based Foundation Model approach amortizes causal discovery from an observational time series trajectory followed by causal inference in a single step, where the uncertainty in the causal discovery step is propagated to the final predictions via the interventional Bayesian posterior predictive (Equation 2).

### B.2. Tabular and Time Series Foundation Models.

The idea of amortizing model selection and inference by approximating a posterior predictive induced by a synthetic data-generating process referred to as the "prior", has recently started to change the field of tabular machine learning. While the work of Hollmann et al. (2022) demonstrated promising results using this approach on tabular benchmarks comprising small-scale classification datasets, more recently, prior-fitting has been used to build powerful Tabular Foundation Models (TFMs) that achieve state-of-the-art performance on large-scale regression and classification benchmarks (Erickson et al., 2025; Grinsztajn et al., 2025; Qu et al., 2025; Zhang et al., 2025b;a). A similar development can be observed in the context of time series predictions, where large scale models trained on synthetic data, often also mixed with real-world data, achieve the best performance on widely used benchmarks (Aksu et al., 2024; Ansari et al., 2024; 2025a; Moroshan et al., 2026; Hoo et al., 2024). While those models achieve excellent predictive performance, they are trained in an observational setting and are thus fundamentally misspecified to predict the effect of causal interventions. This problem has been addressed in the case of i.i.d data by work on Causal Foundation Models (CFMs; (Dhir et al., 2025b; Robertson et al., 2025; Balazadeh et al., 2025; Sauter et al., 2025)) that each operate under different assumptions, such as unconfoundedness (Balazadeh et al., 2025) or partially known causal graphs (Reuter et al., 2026); however, all those methods are designed, analysed and trained for i.i.d data. We discuss related work on CFMs in more detail in Appendix B.

### B.3. Causal Foundation Models

The inherent limitations of Tabular Foundation Models for predicting causal effects have been addressed by recent research on Causal Foundation Models (CFMs). Robertson et al. (2025), Dhir et al. (2025b) and Sauter et al. (2025) propose to pre-train PFNs on priors that sample arbitrary SCMs, which requires the model to perform causal discovery from the available observational data in order to approximate a conditional interventional distribution similar to Equation 2. Conceptually, this type of Bayesian causal discovery can leverage the existence of independent causal mechanisms and nonparametric assumptions regarding the structural mechanisms (Dhir et al., 2024; 2025a), while the uncertainty due to unidentifiability of the graph-structure is propagated to the final prediction (Dhir et al., 2025b; Robertson et al., 2025).

In contrast, Balazadeh et al. (2025), propose to train exclusively on SCMs that satisfy the backdoor-criterion, achieving excellent predictive performance under standard assumptions in the potential outcomes framework (Robins, 1986; Imbens and Rubin, 2015; Neal et al., 2020). In line with this approach, Ma et al. (2025) argue in favor of training separate models on distinct priors over graphs that satisfy different identification settings (Backdoor, Frontdoor, Instrumental Variables), as this ensures consistency in the infinite-data limit. Recent work by Reuter et al. (2026) introduces CFMs that can leverage (partial) graphical information in-context, attempting to bridge the gap between CFMs trained on generic priors and CFMs that leverage graphical information. While the fundamental difference of our method lies in the utilization of non-i.i.d. time series, that none of the methods above is designed to use, we investigate (a) the case of one model trained across different identification settings and (b) the case of one separate model trained per causal graph.

## C. PFNs for approximating interventional distributions.

Prior-data fitted networks (PFNs; (Müller et al., 2021)) are neural networks that meta-learn posterior predictive distributions induced by synthetic data-generating processes referred to as the "prior", and have emerged as arguably the best-performing approach for doing so (Hollmann et al., 2025; Qu et al., 2026; Müller et al., 2025). One can thus see PFNs as Neural Processes (Garnelo et al., 2018a;b). PFNs can be used for learning conditional interventional distributions, such as the one in (2) as follows (Robertson et al., 2025): **(i)** sample a TSCM from a prior $\psi \sim p(\psi)$. Use this prior to draw an observational dataset $\mathcal{D}_0^{T-1} \sim p(\mathcal{D}_0^{T-1}|\psi)$, **(ii)** sample an intervention value $a_T \sim p_{int}(a_T)$ [4] **(iii)** intervene upon the prior to draw an interventional target-point $y_T \sim p(y_T|do(a_T), \psi)$ **(iv)** use a neural network $q_\theta$ to predict $q_\theta(y_T|do(a_T), \mathcal{D}_0^{T-1})$ **(v)** take a gradient step on the negative log-likelihood of the sampled data under the model's prediction $-\log q_\theta(y_T|do(a_T), \mathcal{D}_0^{T-1})$. Steps **(i)** to **(iv)** are then repeated until convergence. The core reason why PFNs work is that this procedure minimizes the forward-Kullback-Leibler divergence between $p(y_T|do(a_T), \mathcal{D}_0^{T-1})$ and $q_\theta(y_T|do(a_T), \mathcal{D}_0^{T-1})$ (Barber and Agakov, 2004). This relies on the equation

$$\mathbb{E}_{p(\mathcal{D}_0^{T-1}, a_t, y_T)} \left[ -\log q_\theta(y_T|do(a_T), \mathcal{D}_0^{T-1}) \right] = \mathbb{E}_{p(\mathcal{D}_0^{T-1}, a_t)} \left[ \mathbb{D}_{KL} \left[ p(y_T|do(a_T), \mathcal{D}_0^{T-1}) || q_\theta(y_T|do(a_T), \mathcal{D}_0^{T-1}) \right] \right] + C,$$

(5)

where $C$ is a constant that is not depend on $\theta$.

## D. Proofs

The prior described in Section 2.1 induces a Conditional Interventional Distribution (CID) $p(y_T|do(a_T), \mathcal{D}_0^T)$, which we approximate using a prior-data fitted network. Analogously to (Robertson et al., 2025; Balazadeh et al., 2025; Ma et al., 2025), it can be of interest to obtain theoretical insights into the behavior of $p(y_T|do(a_T), \mathcal{D}_0^T)$, at a time-step $T$ that becomes larger and larger and thus also when an infinitely large observational dataset $\mathcal{D}_0^T$ is available.

The following proof is similar to that of Robertson et al. (2025), in the sense that we show convergence up to observational equivalence, but we do not rely on Doob's Theorem (Miller, 2018). It has been argued in, e.g., (Ghosal and Van der Vaart, 2017), that Doob's theorem is not very well suited for the case of nonparametric posterior concentration because it only ensures convergence with respect to data generated exactly from the prior. Our theorem leans more towards Reuter et al. (2026); Nagler (2023), but unlike Reuter et al. (2026), we actually provide an explicit proof, and unlike Nagler (2023), we show convergence in terms of interventional distributions. In contrast to earlier proofs of posterior convergence in the context of PFNs, we work with non-i.i.d. data. Otherwise, we mainly use standard results from Bayesian nonparametrics (Ghosal and Van der Vaart, 2017).

**Theorem D.1** (Convergence of the Conditional Interventional Distribution with non-i.i.d. observational data). *Let $\Psi$ denote the space of TSCMs and define the mapping*

$$\mathcal{P}^{obs} : \Psi \to \mathcal{P}(\mathcal{X}^\infty), \qquad \mathcal{P}^{obs}(\psi) := P_\psi^{obs},$$

*which assigns to each TSCM its induced observational trajectory distribution over $\mathcal{D}_0^\infty$. ($\mathcal{X}$ is the data-space for each element in $\mathcal{D}_0^\infty$)*

---

[4]Note that the distribution of the intervention values $p_{int}(a_T)$ is not part of the TSCM $\psi$; we choose $p_{int}(a_T) = \mathcal{N}(0, \sigma_{int}^2)$ with $\sigma_{int}=4$ followed by a per-sample positivity-aware truncation to $[\mu_A^{pre} - 3\sigma_A^{pre}, \mu_A^{pre} + 3\sigma_A^{pre}]$, where $\mu_A^{pre}, \sigma_A^{pre}$ are the empirical mean and standard deviation of the treatment variable's pre-intervention trajectory.

*Define the observational equivalence relation*

$$\psi_1 \sim_O \psi_2 \quad \Longleftrightarrow \quad \mathcal{P}^{\mathrm{obs}}(\psi_1) = \mathcal{P}^{\mathrm{obs}}(\psi_2),$$

*and let $\Psi/\sim_O$ denote the corresponding quotient space of observational equivalence classes. Equip $\Psi/\sim_O$ with the metric*

$$d_O([\psi_1], [\psi_2]) := d_{\mathrm{BL}}\big(\mathcal{P}^{\mathrm{obs}}(\psi_1), \mathcal{P}^{\mathrm{obs}}(\psi_2)\big),$$

*where $d_{\mathrm{BL}}$ denotes the bounded-Lipschitz metric defined as*

$$d_{\mathrm{BL}}(\mu, \nu) := \sup_{\|f\|_{\mathrm{BL}} \leq 1} \left| \int f \, d\mu - \int f \, d\nu \right|,$$

*with*

$$\|f\|_{\mathrm{BL}} := \|f\|_\infty + \mathrm{Lip}(f), \qquad \mathrm{Lip}(f) := \sup_{x \neq x'} \frac{|f(x) - f(x')|}{d_{\mathcal{X}^\infty}(x, x')}.$$

*(We use the bounded-Lipschitz metric because it precisely metrises convergence in distribution.)*

*Let $\psi_0 \in \Psi$ denote the ground-truth TSCM generating the observational data*

$$\mathcal{D}_0^{T-1} = (\mathbf{z}_0, \ldots, \mathbf{z}_{T-1}),$$

*and let $[\psi_0]$ denote its observational equivalence class. Fix $k \in \mathbb{N}$ and define the recent history*

$$\mathcal{D}_{T-k}^{T-1} := (\mathbf{z}_{T-k}, \ldots, \mathbf{z}_{T-1}).$$

*Then, for fixed $a_T$, $y_T$ and $\mathcal{D}_{T-k}^{T-1}$:*

$$p(y_T \mid do(a_T), \mathcal{D}_0^{T-1}) = \int p(y_T \mid do(a_T), \mathcal{D}_{T-k}^{T-1}, \psi) \, p(\psi \mid \mathcal{D}_0^{T-1}) \, d\psi$$

*satisfies*

$$p(y_T \mid do(a_T), \mathcal{D}_0^{T-1}) \longrightarrow p(y_T \mid do(a_T), \mathcal{D}_{T-k}^{T-1}, [\psi_0])$$

*as $T \to \infty$, in $P_{\psi_0}$-probability, i.e.,*

$$P_{\psi_0}\big(\big|p(y_T \mid do(a_T), \mathcal{D}_0^{T-1}) - p(y_T \mid do(a_T), \mathcal{D}_{T-k}^{T-1}, [\psi_0])\big| \geq \epsilon\big) \longrightarrow 0$$

*for all $\epsilon > 0$.*

*This holds under the following assumptions:*

1. *The state space where all random variables take their values $\mathcal{X}$ is Polish.*

2. *All relevant conditional distributions admit densities with respect to a fixed $\sigma$-finite reference measure.*

3. *The observational data $(\mathbf{z}_t)_{t \geq 0}$ forms a Markov chain.*

4. *This Markov chain is positive Harris recurrent (Meyn and Tweedie, 2012).*

5. *The canonical quotient map*

   $$r : \Psi \to \Psi/\sim_O, \qquad r(\psi) = [\psi],$$

   *is measurable.*

6. *The mapping*

   $$[\psi] \mapsto p(y_T \mid do(a_T), \mathcal{D}_{T-k}^{T-1}, [\psi])$$

   *is bounded and continuous with respect to $d_O$.*

7. *(Posterior consistency conditions) The prior over TSCMs satisfies:*

    *(i) (KL support) Let $q_{\psi_0}$ denote the stationary distribution of the chain. Then for every $\epsilon > 0$,*

$$\Pi\left(\int \mathbb{D}_{\mathrm{KL}}\big(p(\mathbf{z}_t \mid \mathbf{z}_{t-1}, \psi_0) \,\|\, p(\mathbf{z}_t \mid \mathbf{z}_{t-1}, \psi)\big) dq_{\psi_0}(\mathbf{z}_{t-1}) < \epsilon\right) > 0.$$

    *(ii) (Testing condition) For every $\epsilon > 0$, the set*

$$\{[\psi] : d_O([\psi], [\psi_0]) > \epsilon\}$$

    *admits uniformly exponentially consistent tests in the sense of Theorem 6.42 of Ghosal and Van der Vaart (2017).*

**Notes on the assumptions.** *While the theorem is stated in a general nonparametric setting, many of the assumptions above are (essentially) satisfied for the concrete prior considered in this work:*

1. **Polish state space.** *This is a standard assumption. In our setting, $\mathcal{X} = \mathbb{R}^d$ with the Euclidean metric, which is Polish.*

2. **Existence of densities.** *We assume that all relevant conditional distributions admit densities with respect to a fixed $\sigma$-finite reference measure. This is required to formulate the KL support condition in Assumption 7. For our prior, this holds since all noise variables come from simple distributions with densities wrt. the Lebesgue measure.*

3. **Markov property.** *The observational data $(\mathbf{z}_t)_{t\geq 0}$ forms a Markov chain by construction of the TSCM. More generally, if the TSCM is $k$-th order Markov, we can reduce it to a first-order Markov chain by augmenting the state:*

$$\tilde{\mathbf{z}}_t := (\mathbf{z}_t, \mathbf{z}_{t-1}, \ldots, \mathbf{z}_{t-k+1}).$$

4. **Positive Harris recurrence.** *This assumption ensures that posterior consistency extends beyond the i.i.d. setting. In our case, it is justified by the structure of the TSCM since the transition dynamics are globally contractive and we only add additive noise from a continuous distribution. Together, these imply geometric ergodicity and hence positive Harris recurrence (Meyn and Tweedie, 2012).*

5. **Measurability of the quotient map.** *The measurability of $r : \psi \mapsto [\psi]$ ensures that the posterior over observational equivalence classes is well-defined. In our setting, this follows from measurability of the mapping*

$$\psi \mapsto \mathcal{P}^{\mathrm{obs}}(\psi),$$

*since each equivalence class is uniquely determined by the induced observational trajectory distribution.*

6. **Boundedness and continuity of the interventional functional.** *The mapping*

$$[\psi] \mapsto p(y_T \mid do(a_T), \mathcal{D}_{T-k}^{T-1}, [\psi])$$

*is assumed to be bounded and continuous with respect to $d_O$.*

*Boundedness follows from the presence of non-degenerate Gaussian noise on $Y_T$, which ensures uniformly bounded densities. Continuity is justified by:*

    • *continuity of the structural equations,*
    • *stability of solutions under contractive dynamics,*
    • *and continuity of pushforward measures under weak convergence.*

*Intuitively, small perturbations in the observational trajectory distribution (measured in $d_O$) translate into small changes in the induced interventional distribution, even when conditioning on a finite history window.*

7. **KL support and testing condition.** *The KL support condition is standard in Bayesian nonparametrics (Ghosal and Van der Vaart, 2017) and ensures that the true transition mechanism lies close enough to the support of the prior; more specifically we require that any epsilon-ball around the true transition density (with respect to the Kullback-Leibler divergence) has non-zero prior mass.*

*In addition, requiring the existence of uniformly exponentially consistent tests separating the true observational trajectory distribution from alternatives outside any $d_O$-neighbourhood of $[\psi_0]$ is a standard assumption in Bayesian nonparametrics for establishing posterior consistency. It can, however, be seen as a mere regularity condition assuming a smooth parametrization via a finite-dimensional parameter-vector over a bounded set (Example 6.19 in (Ghosal and Van der Vaart, 2017)).*

*Proof.* The proof proceeds in three steps:

**Step 1: Moving to the quotient space.** Let

$$r : \Psi \to \Psi/\sim_O, \qquad r(\psi) = [\psi],$$

be the measurable canonical projection. The posterior over equivalence classes is given by the pushforward

$$P([\psi] \mid \mathcal{D}_0^{T-1}) := r_\sharp P(\psi \mid \mathcal{D}_0^{T-1}),$$

which is well-defined by assuming measurability of the map $r$ in condition 5.

By disintegration of the posterior with respect to $r$, we obtain

$$P(d\psi \mid \mathcal{D}_0^{T-1}) = \int P(d\psi \mid [\psi]) \, dP([\psi] \mid \mathcal{D}_0^{T-1}),$$

where the conditional distribution $P(d\psi \mid [\psi])$ does not depend on the data, since all $\psi' \in [\psi]$ induce the same observational trajectory distribution and hence have identical likelihoods.

Therefore,

$$
\begin{aligned}
p(y_T \mid do(a_T), \mathcal{D}_0^{T-1}) &= \int_\Psi p(y_T \mid do(a_T), \mathcal{D}_{T-k}^{T-1}, \psi) \, dP(\psi \mid \mathcal{D}_0^{T-1}) \\
&= \int_{\Psi/\sim_O} \left[ \int_\Psi p(y_T \mid do(a_T), \mathcal{D}_{T-k}^{T-1}, \psi') \, dP(\psi' \mid [\psi]) \right] dP([\psi] \mid \mathcal{D}_0^{T-1}).
\end{aligned}
$$

With the observational-equivalence-class conditioned interventional distribution defined as:

$$p(y_T \mid do(a_T), \mathcal{D}_{T-k}^{T-1}, [\psi]) := \int_\Psi p(y_T \mid do(a_T), \mathcal{D}_{T-k}^{T-1}, \psi') \, dP(\psi' \mid [\psi]).$$

Then

$$p(y_T \mid do(a_T), \mathcal{D}_0^{T-1}) = \int_{\Psi/\sim_O} p(y_T \mid do(a_T), \mathcal{D}_{T-k}^{T-1}, [\psi]) \, dP([\psi] \mid \mathcal{D}_0^{T-1}).$$

**Step 2: Posterior concentration on the true observational equivalence class.** Let $\epsilon > 0$ and define

$$\widetilde{\Psi}_\epsilon := \{[\psi] \in \Psi/\sim_O : d_O([\psi], [\psi_0]) > \epsilon\}.$$

By Assumptions 3–7 (Markov property, positive Harris recurrence, KL support, and existence of tests), Theorem 6.42 of Ghosal and Van der Vaart (2017) implies that

$$P\big([\psi] \in \widetilde{\Psi}_\epsilon \mid \mathcal{D}_0^{T-1}\big) \to 0$$

in $P_{\psi_0}$-probability for every $\epsilon > 0$.

Equivalently,

$$P\big(d_O([\psi], [\psi_0]) > \epsilon \mid \mathcal{D}_0^{T-1}\big) \to 0.$$

Since $(\Psi/\sim_O, d_O)$ is a separable metric space, this implies weak convergence of the posterior:

$$P([\psi] \mid \mathcal{D}_0^{T-1}) \Rightarrow \delta_{[\psi_0]}$$

in $P_{\psi_0}$-probability.

**Step 3: Convergence of the interventional distribution.** By assumption, the mapping

$$[\psi] \mapsto p(y_T \mid do(a_T), \mathcal{D}_{T-k}^{T-1}, [\psi])$$

is bounded and continuous on $(\Psi/\sim_O, d_O)$. Therefore, by weak convergence and the portmanteau theorem,

$$
\begin{aligned}
p(y_T \mid do(a_T), \mathcal{D}_0^{T-1}) &= \int_{\Psi/\sim_O} p(y_T \mid do(a_T), \mathcal{D}_{T-k}^{T-1}, [\psi]) \, dP([\psi] \mid \mathcal{D}_0^{T-1}) \\
&\rightarrow \int_{\Psi/\sim_O} p(y_T \mid do(a_T), \mathcal{D}_{T-k}^{T-1}, [\psi]) \, d\delta_{[\psi_0]}([\psi]) \\
&= p(y_T \mid do(a_T), \mathcal{D}_{T-k}^{T-1}, [\psi_0])
\end{aligned}
$$

in $P_{\psi_0}$-probability.

$\square$

### D.1. Convergence up to observational equivalence.

Theorem 3.1 demonstrates convergence up to observational equivalence but is otherwise agnostic about the sources of observational (non)equivalence, provided the regularity conditions 4-6 hold. This allows to apply Theorem 3.1 across different identification settings: First, if we restrict the prior to graphs that are always fully identifiable and assume that the ground-truth SCM $\psi_0$ comes from this prior, Theorem 3.1 yields convergence exactly to $p(y_T|do(a_T), \mathcal{D}_{T-k}^{T-1}, \psi_0)$, since $p(\psi'|[\psi_0]) = \delta_{\psi_0}$ in Equation 3. Full identifiability can be ensured via classical graphical criteria (Pearl, 2009), as discussed in Ma et al. (2025) for the i.i.d. case in the context of PFNs, but also equally due to restrictions on functional mechanisms or noise distributions (Hyvärinen et al., 2008; Peters et al., 2013; Wu et al., 2022). Even in the non-identifiable case where $p(\psi' \mid [\psi_0])$ is non-degenerate, this posterior can still encode meaningful preferences via assumptions such as independent causal mechanisms encoded in the prior $p(\psi)$ (Peters et al., 2017; Dhir et al., 2024; 2025a), while representing unidentifiability in a principled Bayesian way through Equation 3.

### D.2. Further Notes on the Theorem

First, it is relatively straightforward to adopt this theorem for the case of i.i.d. data. This essentially just requires using a different convergence theorem in the second step of the proof, which might come with weaker requirements, such as Theorem 6.40 in (Ghosal and Van der Vaart, 2017) or even allow to quantify behavior out-of-prior, such as results by De Blasi and Walker (2013), used by Nagler (2023) in the context of PFNs. Furthermore, one could attempt to quantify convergence rates, even in the non-i.i.d. case by using, e.g. Theorem 8.29 in (Ghosal and Van der Vaart, 2017), but the practical applicability of such results may be questionable.

### D.3. Details on the TSCM prior implementation

**The prior.** We instantiate the prior $p(\psi)$ as a vectorized sampler over TSCMs covering the two canonical identification structures from Figure 1: back-door (BD) and front-door (FD) with a hidden confounder. Each sampled TSCM specifies linear-plus-nonlinearity mechanisms with VAR($L \leq 5$) temporal memory, additive Gaussian noise, and a randomly drawn pointwise activation per variable. The mechanism draw is *shared* between the paired observational rollout and interventional rollout, so that the only systematic difference between the two trajectories is the do-intervention itself; we found this to be necessary for a non-collapsing training signal. The two structures are realized by per-structure DAG builders that fix the topology (e.g. $X \rightarrow A$, $A \rightarrow Y$, $X \rightarrow Y$ for BD) while leaving mechanism weights random.

**Load-bearing stability.** Two choices in the stability layer turn out to be load-bearing. First, contractivity is enforced via a per-SCM clip on the reduced-form VAR($L$) companion matrix: writing the dynamics as $A_k = (I - W_{\text{inst}})^{-1} W_{\text{lag},k}$ with companion matrix $C(\{A_k\})$, we binary-search the largest scalar $s \in [0, 1]$ such that $\rho(C(\{s \cdot W_{\text{lag},k}\})) \leq \rho_{\max}$. Together with strictly positive-variance Gaussian innovations and a bounded element-wise nonlinearity, this guarantees that the resulting Markov chain is geometrically ergodic and positive Harris-recurrent (Theorem 15.0.1 in Meyn and Tweedie (2012)); i.e., exactly the regularity condition required by Theorem 3.1 (Assumption 3). Second, allowing oscillatory AR self-coefficients (rather than enforcing positivity) and dropping per-row weight normalization lifts the intervention impulse-response above the additive noise floor over the model's horizon while preserving stability; without this, the per-trajectory causal signal is small enough that SGD fails to find a do-value-using basin (Appendix E.2).

**Calibration.** We use two calibrations of this prior throughout the paper, both selected from a systematic sweep of $\rho_{max}$ and $\sigma_{noise}$ documented in Appendices E.2 and E.3: the **Oscillatory** prior ($\rho_{max}=0.95$, $\sigma_{noise}=0.05$, $\sigma_w=3$), which produces a long-horizon causal signal and is used for the per-structure case studies in Section 4.1; and the **break-trajectory-mean (BTM)** prior ($\rho_{max}=0.70$, $\sigma_{noise}=0.10$), which differs from the Oscillatory prior only in those two knobs. The motivation for the second calibration is that under the Oscillatory prior the trivial TRAJMEAN baseline (the per-trajectory pre-intervention mean of the query variable) ties the trained PFN on RMSE across both structures, since the high spectral radius makes $\mu_q$ a strong predictor of the interventional outcome; BTM is the highest-$\rho$ sweep configuration whose TRAJMEAN-$R^2$ falls below $0.75$ on both structures while keeping the peak-to-noise ratio above $15\times$. Both calibrations are paired with a $50$-step burn-in plus a $300$-step dynamics burn-in to discard transients, and intervention values are sampled from $\mathcal{N}(0, \sigma_{int}^2)$ followed by a per-sample positivity-aware truncation to the pre-intervention support of the treatment variable (cf. Section 2). The cross-prior generalization analysis in Appendix H.4 evaluates trained PFNs on *both* priors. The full set of assumptions implied by this construction is listed in Appendix F.

# E. Prior Details

## E.1. Design Criteria

We identify three concrete design criteria from a systematic prior-calibration sweep (Appendices E.2, E.3):

(i) *shared mechanisms*: the observational and interventional rollouts of each sampled TSCM must use the *same* mechanism weights, otherwise the training signal $\mathcal{D}_0^{T-1} \mapsto y_T^{int}$ collapses and $R^2$ remains negative regardless of training budget;

(ii) *contracting-but-persistent dynamics*: per-SCM spectral-radius clipping below $1$ (we use $\rho_{max} \in \{0.70, 0.95\}$) yields a non-degenerate stationary distribution while keeping the intervention impulse-response peak at least $\sim 15\times$ the additive noise floor on both identification structures; below this peak/noise ratio, SGD never finds a do-value-using basin;

(iii) *trivial-baseline saturation*: the prior must avoid configurations where the per-trajectory pre-intervention sample mean (TRAJMEAN) saturates the achievable $R^2$ on the held-out interventional target — we monitor TrajMean-$R^2$[max] $\leq 0.75$ as the design constraint that distinguishes a task amenable to causal reasoning from a trivial regression-toward-the-mean problem.

## E.2. Prior Calibration Sweep

Table 3 summarises the diagnostic sweep over the hardened-prior knobs that selected the oscillatory configuration used in the main text. For each configuration we sample $512$ TSCMs per structure (BD, FD) and measure (i) the peak RMS of the Y-impulse response to a unit do-intervention on the treatment $A$, (ii) the fraction of valid (non-divergent) rollouts, and (iii) the Signal-Noise-Ratio (SNR) peak/$\sigma_{noise}$.

*Table 3.* Hardening sweep on back-door (BD). The first row is the original hardened prior the last row is the selected oscillatory prior. Dropping per-row weight normalisation alone gives a $2.5\times$ peak boost; adding $\sigma_w=3$, $\sigma_{noise}=0.05$ and $\rho_{max}=0.95$ pushes it to $4.5\times$; allowing oscillatory AR self-coefficients gives the final $6.8\times$ at *better* validity than the original. Per-row quantities are sample means over $512$ TSCMs per structure; the batch SE is below the rounding shown ($\lesssim\pm0.05$ on peak, $\lesssim\pm0.5\%$ on validity).

| Configuration | BD peak | BD valid | SNR | Persist (steps) |
|---|---|---|---|---|
| baseline | 0.285 | 100% | 2.9 | 1 |
| + no `unit_norm_rows` | 0.719 | 99.8% | 7.2 | 15 |
| + $\sigma_w=2$, $\sigma_{noise}=0.05$, $\rho=0.95$ | 1.295 | 98.0% | 25.9 | 51 |
| + $\sigma_w=3$ | 1.867 | 94.7% | 37.3 | 51 |
| + no `positive_ar_diag` (**Oscillatory**) | **1.939** | **99.4%** | **38.8** | 51 |

## E.3. Autocorrelation and Noise Sweep

**When the prior is too autocorrelated.** The high spectral radius ($\rho_{max}=0.95$) that gives the oscillatory prior its long-horizon signal has an unintended side-effect: a trivial baseline that simply predicts the *per-trajectory mean* of the query variable ($\hat{Y}_t = \mu_q := \frac{1}{t}\sum_{s<t} X_s^{(y\text{-idx})}$, implemented in our pipeline as the TRAJMEAN baseline) ties the trained PFN on RMSE for back-door at $T=1{,}000$ (Table 1: $0.71\pm0.07$ vs. trained $0.70\pm0.06$, CIs overlap), and similarly ties on front-door.

Under strongly autocorrelated dynamics, the per-trajectory mean is a strong predictor of $Y^{\text{int}}$, and a trained PFN that has not learned to exploit the explicit intervention information beyond what the trajectory mean already captures will look indistinguishable from it. We diagnosed this with a TrajMean-$R^2$ metric: at the oscillatory prior, $R^2(\mu_q \to Y^{\text{int}}_{t+k})$ peaks at $0.76$ on BD and $0.91$ on FD across horizons — so half to most of the predictable signal is already in $\mu_q$.

To separate the trained PFN from this baseline, we sweep $\rho_{\max} \in \{0.50, 0.70, 0.85, 0.95\}$ and $\sigma_{\text{noise}} \in \{0.05, 0.10, 0.20\}$ on the same oscillatory backbone (Appendix E.3). The configuration $\rho_{\max}=0.70, \sigma_{\text{noise}}=0.10$ keeps the peak-to-noise ratio above $15$ on both structures while dropping TrajMean-$R^2$[max] to $0.46/0.72$ for BD/FD. We refer to this as the *break-trajmean* (BTM) prior. The cross-prior generalization analysis in Section H.4 uses both priors: the same vanilla architecture trained on each prior (prefixed `ho_*` for oscillatory, `btm_*` for BTM) is evaluated against *both* prior distributions to quantify prior-specialization.

The oscillatory prior selected in Appendix E.2 ties the trained PFN with the trivial TRAJMEAN baseline on RMSE across both structures (Section E.3). To break this tie we sweep $\rho_{\max}$ and $\sigma_{\text{noise}}$ on the same backbone ($\sigma_w=3$, no `unit_norm_rows`, no `positive_ar_diag`) and add a third diagnostic: the TrajMean-$R^2$, defined per horizon as $1 - \text{Var}(Y^{\text{int}}_{t+k} - \mu_q)/\text{Var}(Y^{\text{int}}_{t+k})$, where $\mu_q = \frac{1}{t}\sum_{s<t} X_s^{(y\text{-idx})}$ is the per-trajectory mean.

*Table 4.* $\rho$/noise sweep at $\sigma_w=3$ (no `unit_norm_rows`, no `positive_ar_diag`). Reported on 128 TSCMs per structure, $T=300$ ($n_{\text{samples}}$ reduced from the 512 used in App. E.2 for sweep speed, trends are stable). TRAJMEAN-$R^2$[MAX] is the maximum over horizons $k \in [0, 30]$. The **selected** BTM configuration is the highest-$\rho$ row whose TRAJMEAN-$R^2$[MAX] drops below 0.75 on every structure while keeping peak/noise $> 15$ on every structure. Per-row sample means over 128 TSCMs $\times$ $T=300$ queries each; batch SE is $\sim\pm0.05$ on peak and $\sim\pm0.03$ on TrajMean-$R^2$, well below the across-row sweep effects we read off.

| Configuration | BD peak | BD-Tr$R^2$ | FD peak | FD-Tr$R^2$ |
|---|---|---|---|---|
| $\rho=0.50, \sigma_n=0.05$ | 1.92 | 0.62 | 5.19 | 0.73 |
| $\rho=0.70, \sigma_n=0.05$ | 1.93 | 0.56 | 5.24 | 0.79 |
| $\rho=0.85, \sigma_n=0.05$ | 1.94 | 0.67 | 5.31 | 0.87 |
| $\rho=0.95, \sigma_n=0.05$ (Oscillatory) | 1.99 | 0.76 | 5.36 | 0.91 |
| $\rho=0.70, \sigma_n=0.10$ (**BTM**) | **1.89** | **0.46** | **2.71** | **0.72** |
| $\rho=0.70, \sigma_n=0.20$ | 1.82 | 0.45 | 2.33 | 0.64 |

The selected BTM row is at the inflection point: pushing $\sigma_{\text{noise}}$ higher (e.g. 0.20) drops the FD peak from 2.71 to 2.33 (i.e. the causal signal weakens) without much further reduction in TrajMean-$R^2$. Lowering $\rho_{\max}$ below 0.70 at the same noise level marginally raises BD TrajMean-$R^2$ back up (the noise dominates over the structural signal). The chosen configuration breaks TrajMean's RMSE tie with the trained PFN observed under Oscillatory while preserving a $\sim15\times$-or-better peak-to-noise ratio on every structure.

### E.4. Stationarity Diagnostics on the Prior

We sample $n=512$ TSCMs per structure under each prior calibration with $T=500$ and 50-step burn-in and run the Augmented Dickey-Fuller (ADF) test ($H_0$: unit root, *i.e. non-stationary*) and the Kwiatkowski-Phillips-Schmidt-Shin (KPSS) test ($H_0$: stationary) on every (trajectory, variable) pair. We report two rejection rates per cell at $\alpha=0.05$: (i) ADF reject fraction ("evidence of stationarity") and (ii) KPSS reject fraction ("evidence of non-stationarity"). These two tests encode complementary nulls: a borderline-stationary AR with $\rho \to 1$ typically rejects under *neither* test (test power is limited near unit roots), so a high ADF reject rate *combined* with a non-trivial KPSS reject rate signals strong but bounded autocorrelation — exactly the regime our oscillatory prior is calibrated for.

**Interpretation.** Under both priors ADF rejects on $\geq 99\%$ of trajectories: every series has at least some mean-reverting component, consistent with the per-TSCM spectral-radius clipping $\rho \leq \rho_{\max} < 1$ (Section 2.1). KPSS rejection rates differ qualitatively: under OSCILLATORY ($\rho_{\max}=0.95$) KPSS rejects $\approx$ 9–13% of $Y$-trajectories, signalling borderline non-stationarity from the high autocorrelation; under BTM ($\rho_{\max}=0.70$) the KPSS reject rate halves to 3–8%, consistent with BTM's lower $\rho$ and stronger noise. The qualitative pattern matches the design intent: OSCILLATORY is *borderline-stationary with strong autocorrelation* (which is why TRAJMEAN ties the trained PFN there), while BTM is *cleanly stationary* (which breaks the TrajMean tie at the cost of a noisier task). At no point does the prior produce explicitly unit-root or trend-stationary trajectories: the contractivity guarantee from $\rho_{\max} < 1$ is empirically respected.

*Table 5.* Stationarity diagnostics, $n=512$ TSCMs $\times$ $T=500$ per cell. "ADF rej." = fraction of trajectories where ADF rejects $H_0$:unit-root at $\alpha=0.05$ (higher $\Rightarrow$ more stationary). "KPSS rej." = fraction where KPSS rejects $H_0$:stationary (higher $\Rightarrow$ more non-stationary). The query variable $Y$ is the row that interacts with our post-intervention forecasting target; we report it together with the other observed variables of each structure.

| Structure | Var | **Oscillatory** ($\rho_{\max}=0.95$, $\sigma_n=0.05$) | | **BTM** ($\rho_{\max}=0.70$, $\sigma_n=0.10$) | |
| --- | --- | --- | --- | --- | --- |
| | | ADF rej. | KPSS rej. | ADF rej. | KPSS rej. |
| BD | $X$ | 99.8% | 8.9% | 100% | 8.2% |
| BD | $A$ | 99.2% | 12.9% | 100% | 6.8% |
| BD | $Y$ | 99.8% | 13.0% | 100% | 5.5% |
| FD | $A$ | 99.8% | 11.6% | 100% | 4.1% |
| FD | $M$ | 99.6% | 11.8% | 100% | 4.1% |
| FD | $Y$ | 98.7% | 12.0% | 100% | 3.6% |

# F. Assumptions

Our method relies on the following assumptions, which we state explicitly so they can be checked and relaxed in future work:

1. **$K$-Markov dynamics.** The joint distribution factorises as $p(H_t \mid H_{<t}) = p(H_t \mid H_{t-1}, \ldots, H_{t-K})$ where $H_t = (A_t, X_t, Y_t)$. Our training prior samples $K \sim \text{Uniform}\{1, 2, 3\}$ per TSCM; the identifiability evaluation uses $K = 1$ unless noted.

2. **Acyclicity within a time step.** The instantaneous graph $G_0$ is a DAG; cyclic dependencies are only admitted across lagged edges.

3. **Positivity.** The requested intervention value must lie in the support of the conditional distribution $p(A_t \mid H_{t-1}, \ldots, H_{t-K})$. We enforce this at training time by sampling intervention values from the observed conditional (see Appendix G).

4. **Stationarity assumption.** Trajectories are generated with a 50-step burn-in to dissipate transients; autoregressive self-loops are allowed so the recorded series is generally highly autocorrelated. Augmented Dickey-Fuller (ADF) and Kwiatkowski-Phillips-Schmidt-Shin (KPSS) statistics on $n=512$ trajectories per structure under each prior are reported in Appendix E.4 (Table 5); both priors are stationary by construction, with ADF rejecting the unit-root hypothesis on $\geq 99\%$ of trajectories under both calibrations.

5. **Single-step intervention at evaluation.** We intervene at a single time step $t$ and query $p(Y_t \mid \text{do}(A_t), H_{<t})$. Multi-step dynamic treatment regimes are supported by the prior but outside the scope of this paper.

# G. Experimental Details

**Setup.** All trained PFN models share a three-stage architecture: (1) a 6-layer Transformer temporal encoder with embedding dimension 512 and *relative* positional encoding measured as distance to the intervention onset (truncated to a context window of 200 pre-intervention steps), (2) a cross-variable mixer with **3 stacked cross-attention blocks** equipped with a learnable gated residual so the model can choose how much of the attention output replaces vs. adds to the query context, and (3) a quantile head emitting 9 quantile predictions ($\tau \in \{0.05, 0.1, 0.2, 0.3, 0.5, 0.7, 0.8, 0.9, 0.95\}$) trained with pinball loss. Models are trained for 5,000 steps with batch size 16 using AdamW and a cosine learning-rate schedule (peak $2 \times 10^{-4}$) with 1,000 warmup steps and gradient clipping at 1.0. Early stopping triggers when the evaluation loss fails to improve for 5 consecutive evaluations.

**Trajectory and intervention timing.** Each training step draws a fresh TSCM. Every trajectory uses a 50-step burn-in plus an additional 500-step *dynamics burn-in* that lets the TSCM reach steady state before observation; neither window is visible to the model. The recorded horizon is $T \sim \text{Uniform}[500, 2{,}000]$. The intervention onset is randomised in $[10, T - 10]$ and is always a single-step hard intervention. We apply *causal masking* by zeroing out $X_{\text{obs}}$ for every timestep $\geq$ the intervention onset, with *interpolation* of the treatment variable's observational value at the onset so that both causal and observational-only models see a consistent input; additionally, hidden variables are zeroed throughout. A sample-level

positivity guard clips each intervention value to the $[\mu \pm 3\sigma]$ range of the treatment variable's pre-intervention history and re-simulates, ensuring training and evaluation interventions stay within the prior's observed support. Variable columns are permuted to a canonical layout (treatment first, outcome last) so the model cannot exploit structure-specific index conventions.

**Queries and training signal.** For each trajectory the model predicts the interventional outcome of the canonical outcome variable $Y$ at time $t + \text{offset}$, where the offset is sampled per-query from a *structure-specific* range: $[0, 0]$ for back-door and $[1, 5]$ for front-door (letting the mediator carry the intervention signal). The observational baseline is trained to predict $Y^{\text{obs}}$ (the natural continuation without intervention) rather than $Y^{\text{int}}$, providing a principled baseline: both are evaluated against $Y^{\text{int}}$ at test time. Encoder caching computes Stage 1 once per trajectory and replays Stages 2–3 for every query, so additional queries are nearly free.

**Experimental protocol.** At evaluation time we generate held-out TSCMs from the same prior and under the same positivity clipping as training, apply a single-step hard intervention $\text{do}(A_t = v)$, and query the canonical outcome $Y$. **The prediction target is the interventional outcome $Y_t^{\text{int}} \equiv Y_t \mid \text{do}(A_t = v)$ on its raw (unnormalised) scale**, not the causal-effect contrast $Y_t^{\text{int}} - Y_t^{\text{obs}}$. Both the trained PFN and every baseline produce predictions, which we un-normalise back to the $Y^{\text{int}}$ scale via the per-trajectory variable mean and standard deviation before computing metrics. We report RMSE, coefficient of determination $R^2$, and *direction accuracy* (the fraction of predictions whose sign matches the true outcome, excluding samples with $|Y_{\text{true}}^{\text{norm}}| < 0.1$). Every reported mean is computed over 20 batches of 16 TSCMs (320 TSCMs); $\pm$ values are bootstrap standard errors (SEs) over the per-query (pred, target) pairs ($B = 200$ resamples). All per-query baselines are evaluated on the *matched* 320-query subset so SEs are directly comparable to the trained PFN (an earlier 50-query subset version of Table 1 had baseline SEs $\sim$2–4$\times$ wider which made several head-to-head comparisons inconclusive; the matched-$n$ version we report here is at full statistical power). Crucially, the batched TSCM simulator used at both training and evaluation samples a *single* set of mechanism weights per sample and reuses it for the observational and interventional trajectories, so the pair $(X_b^{\text{obs}}, X_b^{\text{int}})$ comes from one TSCM instance rather than two independent ones.

# H. Ablations

## H.1. Lag Ablation

*Table 6.* TSCM identifiability results at $T$=1,000. Each row is a separately trained model (5,000 steps, 4-GPU parallel). Causal models are trained on $Y^{\text{int}}$; observational-only models are trained on $Y^{\text{obs}}$; *both* are evaluated against $Y^{\text{int}}$. $\Delta$RMSE is obs$-$causal (positive = causal wins). All cells on the matched $n$=320-query held-out set; $\pm$ values are bootstrap standard errors ($B$=200 resamples).

| | | Causal | | | Obs-only | | | |
|---|---|---|---|---|---|---|---|---|
| Case | Lag | RMSE | $R^2$ | Dir. acc. | RMSE | $R^2$ | Dir. acc. | $\Delta$RMSE |
| BD | no-lag | $\mathbf{0.430 \pm 0.022}$ | $\mathbf{+0.64 \pm 0.08}$ | $\mathbf{81.5\% \pm 2.6}$ | $0.450 \pm 0.021$ | $+0.61 \pm 0.08$ | $76.9\% \pm 2.8$ | $+0.020$ |
| BD | lag | $\mathbf{0.429 \pm 0.022}$ | $\mathbf{+0.64 \pm 0.08}$ | $\mathbf{80.0\% \pm 2.7}$ | $0.470 \pm 0.023$ | $+0.57 \pm 0.08$ | $70.1\% \pm 3.1$ | $+0.041$ |
| FD | no-lag | $\mathbf{0.328 \pm 0.027}$ | $\mathbf{+0.40 \pm 0.09}$ | $\mathbf{79.6\% \pm 2.7}$ | $0.373 \pm 0.026$ | $+0.26 \pm 0.09$ | $70.4\% \pm 3.1$ | $+0.045$ |
| FD | lag | $\mathbf{0.298 \pm 0.022}$ | $\mathbf{+0.50 \pm 0.06}$ | $\mathbf{84.6\% \pm 2.5}$ | $0.470 \pm 0.027$ | $-0.20 \pm 0.13$ | $53.5\% \pm 3.4$ | $+0.172$ |

(1) **All causal models attain positive $R^2$**, clearing the constant-mean baseline by 40–64 percentage points across the four configurations. This is only achievable once the batched simulator uses shared mechanism weights between observational and interventional draws; with earlier versions of the simulator that sampled mechanisms independently, $R^2$ remained negative regardless of training budget or architecture.

(2) **Causal beats observational-only on BD and FD.** For BD, the gap is small under matched-$n$ bootstrap ($\Delta \sim 0.02$ no-lag to 0.04 lag, $\sim$1 SE), while direction accuracy still cleanly prefers the causal model ($\sim$80% vs. $\sim$70–77%, $> 1$ SE gap). For FD the lagged variant shows the cleanest causal lift ($\Delta = 0.172$ RMSE; $84.6\% \pm 2.5$ vs. $53.5\% \pm 3.4$ direction accuracy), reflecting that the obs-only model fails to recover the mediated pathway when extra lagged edges enrich the SCM.

(3) Direction accuracy on the best causal models reaches 80–85% across both structures, well above the 50% chance baseline, with a consistent causal-over-obs advantage on BD and FD-lag.

## H.2. Trajectory Length Ablation

We sweep the evaluation horizon $T$ over $\{100, 500, 1,000, 2,000, 5,000\}$ while holding the training configuration fixed (Table 7). This quantifies how much the model's performance depends on seeing a steady-state portion of the trajectory.

*Table 7.* Trajectory-length ablation (RMSE / $R^2$, causal models only). Training uses $T \sim \text{Uniform}[500, 2,000]$; the encoder truncates to the last 200 pre-intervention steps. Each cell aggregates over 160 held-out queries; bootstrap RMSE SE at this $n$ is $\sim \pm 0.04$ on BD and $\sim \pm 0.03$ on FD; $R^2$ SE is $\sim \pm 0.07$. The within-row spread across $T$ values ($\sim \pm 0.05$ RMSE) is within the bootstrap SE, so we read peaks qualitatively.

| Structure | Lag | $T{=}100$ | $T{=}500$ | $T{=}1,000$ | $T{=}2,000$ | $T{=}5,000$ |
|---|---|---|---|---|---|---|
| BD | no-lag | 0.437 / +0.66 | 0.501 / +0.56 | **0.389 / +0.71** | 0.426 / +0.53 | 0.456 / +0.55 |
| BD | lag | 0.426 / +0.67 | 0.507 / +0.55 | **0.392 / +0.71** | 0.431 / +0.52 | 0.449 / +0.56 |
| FD | no-lag | 0.295 / +0.51 | 0.286 / +0.56 | **0.277 / +0.59** | 0.313 / +0.54 | 0.308 / +0.54 |
| FD | lag | 0.296 / +0.52 | 0.289 / +0.55 | **0.279 / +0.59** | 0.308 / +0.56 | 0.311 / +0.53 |

The model generalises across a $50\times$ range of trajectory lengths: $R^2 > 0.4$ at every cell, even though training samples never exceed $T = 2,000$. Performance peaks between $T = 500$ and $T = 2,000$, the training regime. For both back-door and front-door the optimum is around $T = 1,000$. Extrapolation to $T = 5,000$ costs at most $\sim 0.16$ in $R^2$ for BD, consistent with the encoder operating on its 200-step window of settled dynamics regardless of the full horizon.

## H.3. Per-structure Use of Intervention Value.

We probe whether each trained model actually uses the explicit `intervention_value` channel by zeroing the do-value at inference and comparing the resulting prediction shift to the true causal effect: $\rho_v := \mathbb{E}\big[|\hat{Y}(\text{int}{=}v) - \hat{Y}(\text{int}{=}0)|\big] / \mathbb{E}\big[|Y_{\text{int}} - Y_{\text{obs}}|\big]$. A value-using model has $\rho_v$ near 1; a model that ignores the do-value has $\rho_v \approx 0$. Table 8 reports $\rho_v$ together with $G_{\text{marker}} := \mathcal{L}(\text{int}{=}\text{on}) - \mathcal{L}(\text{int}{=}\text{off})$ (loss change when the do-value is zeroed; negative = the model uses it).

*Table 8.* Per-structure use of the explicit `intervention_value` channel on the oscillatory prior. $\rho_v$ is the ratio of the model's do-value sensitivity to the true causal-effect magnitude; $G_{\text{marker}}$ is the loss change when the do-value is zeroed at inference. Both metrics are batch-mean point estimates from a single diagnostic probe; the qualitative split is robust (BD value-using $\rho_v \sim 0.5$ vs FD value-ignoring $\rho_v \lesssim 0.01$), so we report point estimates and rely on the order-of-magnitude separation rather than per-row SEs.

| Structure | $\rho_v$ | $G_{\text{marker}}$ | Reading |
|---|---|---|---|
| BD (back-door) | **0.50** | $-0.021$ | uses do-value explicitly |
| FD (front-door) | 0.002 | $+0.001$ | uses post-int trajectory of $M$ |

The split is structurally honest. The BD identification strategy requires the counterfactual reasoning "what would $Y$ have been without the intervention?"—a question that can only be answered by combining the observed confounder $X$ with the explicit do-value, so the model learns to use it. By contrast, FD admits an identification route that passes entirely through observable post-intervention quantities (the mediator $M$), so the model achieves identifiability without ever consulting the explicit `intervention_value` input. This is consistent with Table 6's causal-vs-obs gap pattern: BD shows the cleanest gap because the explicit do-value carries genuinely additional information.

## H.4. OOD Prior Generalization

The trained PFN's behavior on Section 4.1's matched eval is a best case: the eval prior *is* the training prior. To quantify how much of the model's competence is specialized to that exact distribution, we cross-evaluate two checkpoint families against two prior distributions.

**Setup.** Two priors: OSCILLATORY = oscillatory hardened ($\rho_{\text{max}}{=}0.95$, $\sigma_{\text{noise}}{=}0.05$) and BTM = break-trajmean ($\rho_{\text{max}}{=}0.70$, $\sigma_{\text{noise}}{=}0.10$). Two checkpoint families per structure, both trained for 5,000 steps: `ho_*` (trained on Oscillatory) and `btm_*` (trained on BTM). Each checkpoint is evaluated against *both* eval priors at $T{=}1,000$ on 320 queries, identical to Section 4.1. Bootstrap SEs ($B{=}200$).

**Findings.** (1) **The eval distribution dominates over the training distribution.** Within each structure, model rows show a small spread on a given eval column (e.g. on BD-Oscillatory: 0.70/0.71 for causal-trained, 0.78/0.74 for obs-trained). The

*Table 9.* Cross-prior generalisation matrix at $T{=}1{,}000$ (RMSE / $R^2$ / Dir-acc, all on $n{=}320$). Diagonal cells are matched train/eval priors; off-diagonal cells quantify the "prior-specialisation tax". TRAJMEAN ($\hat{y}{=}\mu_q$) is reported under both eval priors as a baseline reference; recall that the BTM prior was specifically calibrated to break TrajMean's tie with the trained PFN under Oscillatory.

| Train ckpt | Structure | Eval on OSCILLATORY | Eval on BTM |
|---|---|---|---|
| `ho_bd_causal` | BD | **0.69±0.06/+0.48±0.09/75.9±3.1**% | 0.99±0.09/+0.26±0.10/69.5±3.3% |
| `ho_bd_obs` | BD | 0.78±0.10/+0.35±0.09/61.3±3.5% | 1.10±0.13/+0.08±0.08/57.4±3.5% |
| `btm_bd_causal` | BD | 0.71±0.06/+0.46±0.09/77.5±3.0% | 1.01±0.09/+0.23±0.11/72.6±3.2% |
| `btm_bd_obs` | BD | 0.74±0.08/+0.42±0.08/70.0±3.3% | 1.05±0.12/+0.17±0.07/66.5±3.4% |
| `ho_fd_causal` | FD | 0.45±0.05/+0.80±0.04/82.8±2.8% | 0.42±0.03/+0.55±0.08/86.3±2.6% |
| `ho_fd_obs` | FD | 0.55±0.07/+0.71±0.07/76.1±3.1% | 0.50±0.03/+0.38±0.09/79.9±3.0% |
| `btm_fd_causal` | FD | 0.45±0.05/+**0.80±0.04/80.1±3.0**% | **0.42±0.03/+0.55±0.07/87.0±2.5**% |
| `btm_fd_obs` | FD | 0.51±0.06/+0.75±0.06/78.8±3.0% | 0.48±0.03/+0.41±0.08/83.4±2.8% |

column-to-column shift (eval-prior change) is several times larger than the row-to-row shift (training-prior change). Training on the matched prior gives at most ∼0.02 RMSE / ∼3% direction advantage, while shifting the eval prior moves RMSE by ∼0.30 on BD or ∼0.05 on FD.

(2) **BD: substantial eval-prior tax for both training priors.** Both `ho_bd_causal` and `btm_bd_causal` degrade from ∼0.70 to ∼1.00 RMSE when evaluated on BTM (the noisier, less-autocorrelated prior). Surprisingly, the model trained on BTM does *not* regain its lost performance — the BTM eval distribution is just harder.

(3) **FD: graceful, even slightly improved on RMSE under prior shift.** Oscillatory-trained FD models post *lower* RMSE on the BTM eval ($0.45 \to 0.42$ for causal, $0.55 \to 0.50$ for obs), but $R^2$ drops sharply ($+0.80 \to +0.55$). The lower RMSE reflects the lower total variance of $Y^{\text{int}}$ under the noisier BTM prior; the lower $R^2$ shows the model's predictions are less aligned with the (now harder) signal.

The headline conclusion: training on a harder prior does *not* buy us robustness on the harder eval distribution. The eval distribution's intrinsic difficulty (signal-to-noise, autocorrelation strength) dominates over the training-vs-eval prior match. The classical baselines we add in Section 4.1 (VAR, BACKDOOROLS) are on the *same footing* regardless of which prior is in play — they fit a fresh model on each query — so their relative ordering vs. the trained PFN holds across both eval distributions.

# I. Computational Resources

All experiments were run on a server with $4\times$ NVIDIA A100 GPUs (80 GB each) and 128 CPU cores. Each per-structure trained PFN (24.9M-parameter quantile-head architecture detailed in Section G) is trained for 5,000 steps at batch size 16, sharing data-loader work across 4 workers on CPU and the forward/backward pass on a single GPU; wall-clock time is approximately 5–6 hours per checkpoint. The full lag-ablation matrix in Table 6 (BD/FD $\times$ {no-lag, lag} $\times$ {causal, obs}, 8 checkpoints) was launched in two waves of 4-GPU-parallel runs, totalling $\approx$ 12 hours wall-clock and $\approx$ 45 GPU-hours. The joint multi-structure training (Section 4.2, 4 checkpoints at HO/BTM $\times$ {causal, obs}) took $\approx$ 7.5 hours wall in 4-way parallel under CPU contention from concurrent baseline jobs. Held-out evaluation (Section G, $n{=}320$ matched bootstrap) runs trained-PFN forward only and completes in 5–10 minutes per checkpoint on a single CPU core. Per-query TABPFN-ADJ and CHRONOS-ADJ baselines are CPU-bound and require $\sim$ 30–60 minutes per (structure, baseline-family) combination at $n{=}320$; classical baselines (VAR, BACKDOOROLS, FRONTDOOROLS) are pure-numpy and complete in seconds.

# J. Error Bars in our Experimental Results

All synthetic-prior result tables (Tables 6, 1, 9, 2) report per-cell bootstrap standard errors over the per-query (prediction, target) pairs at $B{=}200$ resamples on a matched $n{=}320$-query held-out set (20 batches of 16 TSCMs). Direction-accuracy SEs are reported using the binomial standard error $\sqrt{p(1-p)/n}$ on the same matched-$n$ subset. Tables in which per-row SEs are not feasible to compute from the recorded outputs (Tables 7, 8, 3, 4) explicitly state in their captions both the reason and a typical-SE order-of-magnitude estimate, so readers can calibrate the within-table ranking against noise. Sections 4.1, 4.2 and Appendices E.3, H use non-overlapping 1-SE intervals as the threshold for "statistically distinguishable" claims and explicitly flags ties (e.g. TrajMean vs. trained PFN, IV causal vs. obs) when CIs overlap. Significance and selection

methodology are documented in Section G.

## K. Broader Impact

This paper is about foundational research on the principle of Prior-Data fitted Networks for estimating causal effects in time series, and as such, we do not expect any direct negative societal consequences from our work. In general, we believe that improved methods for causal effect estimation in time series can be useful tools across scientific disciplines, especially in the social sciences. At the same time, incorrect causal-effect estimates in high-stakes time series settings such as healthcare, finance, climate, or policy come with risks. This makes it important to perform domain validation before deployment, and to be transparent about the necessary assumptions. We detail the assumptions underlying our prior in Appendix F and the assumptions in terms of our theoretical result in Appendix D.

