# OpenReview forum: "Causal Foundation Models for Time Series based on Prior-Data fitted Networks"
_ICML.cc/2026/Workshop/FMSD — FMSD @ ICML 2026 Poster_

### Official Review · Reviewer_opVN · 2026-05-20
**Time Series Causal Foundation Models using Temporal Structural Causal Models**

**Rating:** 8
**Confidence:** 2

**Review:**

# Summary

This paper develops an approach for extending Causal Foundation Models to the time series domain using Temporal Structural Causal Models. The authors provide theoretical results on the asymptotic behavior of the conditional interventional distribution induced by a TSCM prior. They empirically evaluate PFN-trained models showing interventionally trained models outperform observational-only models in temporal causal inference.

# Strengths

* Strong theoretical soundness under assumptions for CID convergence
* Empirical results indicate that PFNs benefit from interventional TSCM pretraining over observational-only training
* Extensive ablation and secondary experiments further enforce casual benefits of intervention-trained models.

# Areas for Improvement \+ Detailed Comments

* Missing full break-trajmean results comparing trained PFNs to baseline models. Paper would benefit from inclusion of a copy of Table 1 on BTM in the Appendix.
* Authors note that TrajMean results on OSC indicate that it may not be an optimal benchmark. Inclusion of results from Table 2 and lagged-effects ablation indicate a clear difference in causal and observational models.

---

### Official Review · Reviewer_z7vV · 2026-05-21
**Assessing PFNs for Causal Effect Estimation in Time Series with Temporal Causal Priors**

**Rating:** 8
**Confidence:** 3

**Review:**

## **Summary of contributions**

The paper extends the PFN framework to causal effect estimation in time series, moving beyond standard forecasting objectives. The authors introduce a temporal causal prior based on temporal structural causal models, designed to generate stable trajectories with meaningful causal effects under both back-door and front-door identification settings. The work investigates whether PFNs can effectively estimate causal effects in temporal scenarios, using both observational and interventional data. Experiments compare PFNs against classical baselines and foundation models, investigate whether incorporating interventional information during training and/or during inference is beneficial for causal effect estimation in time series, and evaluate both structure-specific and jointly trained PFNs across causal settings.
The experimental results suggest that incorporating interventional information during training and inference improves performance, particularly in the back-door setting. Moreover, jointly trained models do not degrade performance and can match the results of structure-specific specialist models, indicating that multi-setting training is both robust and effective.


## **Strengths and weaknesses**
### **Strengths**

- Interesting extension of PFNs toward causal effect estimation in time series.
- Clear motivation and well-defined causal prior design.
- Good experimental analysis across back-door and front-door settings.
- The paper is well written and addresses a topic of clear interest to the community. The proposed method is clearly explained, and the experimental setup is clear.

### **Weaknesses**
- Evaluation is limited to controlled synthetic settings, although this limitation is acknowledged by the authors.

## **Suggestions**
- It would be helpful to clarify why simple heuristic baseline performs strongly in the front-door setting compared to adjusted foundation models, whether this is due to TrajMean being better suited to the task, or adjusted foundation models are not well suited to this setting or the evaluation setup.

---

### Official Review · Reviewer_u3iC · 2026-05-21
**An important step towards modeling effective priors for prior-fitted causal models**

**Rating:** 7
**Confidence:** 4

**Review:**

**Summary**

This paper focuses on causal interventional modeling. Contrary to forecasting which models $p(y_t| y_{t-1, …}, x_{t, …})$, the goal is to obtain $p(y_t| y_{t-1, …}, do(x_{t}),  x_{t-1, …})$, where past observed data is generated by a Temporal Structural Causal Model (TSCM), which assigns causal edges between variables and their past values. Intervention corresponds to ignoring parent edges and setting a fixed value at the corresponding node. Since prior-fitted networks (PFNs) have achieved state-of-the-art performance in forecasting, the authors study how to adapt PFN training to this specific task.

Theoretically, multiple causal models can generate the same observations. The authors prove that under certain regularity assumptions, the conditional distribution induced by a prior over TSCMs converges to the distribution averaged over the observational equivalence class of the true TSCM.

Choosing a prior implies choosing how to sample TSCMs. The authors propose a synthetic generator which mainly samples backdoor and frontdoor scenarios, with additional autoregressive dynamics and spectral-radius constraints to enforce stability. This simple prior allows to study the capability of PFNs to model typical causal relationships.

Experimentally, their results show that their PFN trained on their synthetic prior is on par with forecasting foundation models baselines which are explicitly given the causal relationship, and that training on interventional data (TrainedPFNint) outperforms training solely on data without interventions (TrainedPFNobs).

**Strengths**

The paper is well written and includes many details on the experiments. The empirical experiments show that TrainedPFNint outperforms TrainedPFNobs, which highlights the relevance of their approach to construct a prior for causal modeling. They also show the relevance of training jointly on multiple causal scenarios. Appendices include noteworthy proofs and ablation studies.

**Areas for improvement**

Details could be given on the Foundation Models (FMs) baselines. Also, the benchmark is done on very simple causal relationships, which naturally don’t account for most existing realistic scenarios. Furthermore, it seems the forecasting task is fairly easy, as the baseline "repeat the average" (TrajMEAN) itself has impressive results.

**Detailed comment**

For the FM baselines, being “given the correct causal adjustment strategy” is not clear and subject to interpretation. From what I understand (but I may be wrong): the FMs are given as covariates the synchronized series that are implied in the causal graph? (e.g if $Y_t=f(Y_{t-1}) + f(X_{t-2})$, they are given $(Y_{t}), (Y_{t-1}), (X_{t-2})$ ?)

Also, I wonder what is the role of the directional accuracy. Since you are evaluating on $Y_{int}$ and not $Y_{int}-Y_{obs}$, correctly predicting the sign of the output does not seem relevant? I also noticed the furthest relationship to the past is K=3. Perhaps looking at TSCMs the model longer-range temporal dependancies could be interesting. The fact that TrajMEAN is competitive shows the task is too simple. Perhaps to intervention is not strong enough.

Finally, I would have included example plots of generated time series.

**Justification of score**
Overall solid paper, which takes an important step towards modeling effective priors for prior-fitted causal models.